# How the COVID-19 Pandemic Influenced HIV Care: Are We Prepared Enough for Future Pandemics? An Assessment of Factors Influencing Access, Utilization, Affordability, and Motivation to Engage with HIV Services amongst African, Caribbean, and Black Women

**DOI:** 10.3390/ijerph20116051

**Published:** 2023-06-05

**Authors:** Emily McKay, Emmanuela Ojukwu, Saima Hirani, Tatiana Sotindjo, Ijeoma Okedo-Alex, Patience Magagula

**Affiliations:** 1School of Nursing, Faculty of Applied Sciences, University of British Columbia, Vancouver, BC V6T 2B5, Canada; emckay06@student.ubc.ca (E.M.); saima.hirani@ubc.ca (S.H.); 2Department of Pediatrics, Faculty of Medicine, University of British Columbia, Vancouver, BC V6T 2B5, Canada; tsotindjo@cw.bc.ca; 3B.C. Women’s Hospital & Health Centre, Vancouver, BC V6H 2N9, Canada; 4School of Population and Public Health, Faculty of Medicine, University of British Columbia, Vancouver, BC V6T 2B5, Canada; ijeoma11@student.ubc.ca; 5Afro-Canadian Positive Network of BC, Surrey, BC V3T 4H4, Canada; patience.acpnet@gmail.com

**Keywords:** access, utilization, motivation, engagement, affordability, social determinants of health, HIV, AIDS, women’s health, transgender health, healthcare services

## Abstract

The COVID-19 pandemic resulted in disruption in healthcare delivery for people living with human immunodeficiency virus (HIV). African, Caribbean, and Black women living with HIV (ACB WLWH) in British Columbia (BC) faced barriers to engage with HIV care services prior to the COVID-19 pandemic that were intensified by the transition to virtual care during the pandemic. This paper aims to assess which factors influenced ACB WLWH’s access to, utilization and affordability of, and motivation to engage with HIV care services. This study utilized a qualitative descriptive approach using in-depth interviews. Eighteen participants were recruited from relevant women’s health, HIV, and ACB organizations in BC. Participants felt dismissed by healthcare providers delivering services only in virtual formats and suggested that services be performed in a hybrid model to increase access and utilization. Mental health supports, such as support groups, dissolved during the pandemic and overall utilization decreased for many participants. The affordability of services pertained primarily to expenses not covered by the provincial healthcare plan. Resources should be directed to covering supplements, healthy food, and extended health services. The primary factor decreasing motivation to engage with HIV services was fear, which emerged due to the unknown impact of the COVID-19 virus on immunocompromised participants.

## 1. Introduction

In Canada, it is estimated that roughly 63,000 people live with HIV [1,2]. In a cross-sectional survey of 1380 Ontarians, the prevalence of HIV amongst ACB people was found to be five times higher than the actual percentage of ACB people living in Canada [3]. Of the ACB people impacted, about 25% are women [1]. Racialized groups are disproportionately represented in populations living with HIV and face barriers to treatment and engagement with HIV care services [1]. African, Caribbean, and Black women living with HIV (ACB WLWH) face significant barriers to accessing and utilizing HIV care services. HIV care services can be costly and can impact an individual’s motivation to engage in HIV care services [4]. The COVID-19 pandemic has had a substantial impact on ACB WLWH socially, psychologically, economically, and physically [5]. These impacts may influence ACB WLWH’s ability to access, utilize, and afford HIV care services.

ACB communities have also been significantly impacted by the COVID-19 pandemic [5]. Poteat et al. (2020) describe how the systemic impact of racism and various social determinants of health, such as socioeconomic status, have intensified the impact of COVID-19 for the ACB population in a commentary piece [5,6,7]. Etienne et al. (2020) conducted an observational prospective monocentric study of 54 people living with HIV (PLWH) in France to better understand associated factors with severe cases of COVID-19. Of the identified factors associated with increased risk of critical cases of COVID-19, originating from sub-Saharan Africa and older age hold relevant implications for our study sample, ACB WLWH with ages ranging from 21 to 71 [6]. Shiau et al. (2020) published relevant field notes at the beginning of the COVID-19 pandemic, describing the experience of HIV and COVID-19 as a syndemic [7]. It is essential to consider the intersecting impact of COVID-19, HIV, racism, sexism, and gender discrimination and its impact on ACB WLWH within the context of accessing HIV care services. ACB Americans were found to have higher mortality rates from COVID-19 because they are more likely to have chronic health conditions [5,8]. ACB Americans are also more likely to be affected by housing insecurity, multigenerational housing, and employment in essential services, therefore increasing their chances of acquiring COVID-19 [8,9]. In review of the initial stages of the COVID-19 pandemic, Gwadz et al. (2021) published a mixed-methods study that included 96 PLWH. The results of this study demonstrated that HIV care services were frequently disrupted but overall treatment adherence did not seem to falter [8]. In addition, participants described inadequate technological tools for telehealth, a direct impact of structural racism that was intensified by the COVID-19 pandemic [8]. Kullar et al. (2020) published a perspective piece that discussed the way that the COVID-19 pandemic revealed how systemic racism has created unique health disparities, such as increased COVID-19 morbidities and mortalities, for ACB Americans [9]. The added burden and stigmatization of COVID-19 on ACB WLWH may also act as a barrier for women to access, utilize, afford, or to be motivated to engage with HIV care services [7]. The disproportionate impact of COVID-19 on the ACB population demonstrates the need for interventions that address the multifaceted impacts of the social determinants of health.

ACB WLWH are predisposed to a significant number of psychosocial vulnerabilities and are often stigmatized due to their HIV status. ACB PLWH reported increased stigmatization in healthcare settings during the COVID-19 pandemic [10]. Self-stigmatization was also reported in a study, which indicated a connection between a negative self-conceptualization and internalized stigmatization based on HIV status [11]. A cross-sectional survey of 173 ACB WLWH in Ontario was conducted in 2013 to study the relationship between racism, sexism, and psychosocial vulnerabilities [11]. Socio-economic disadvantages, housing insecurity, and psychological stressors also contribute to experiences of stigmatization and risk for treatment non-adherence [11]. Lacombe-Duncan et al. (2020) report the impact of internalized stigma, psychological stressors, and the social determinants of health for transgender WLWH based on a national community-based study in the United States. Their study demonstrates a clear correlation between the prevalence of stigmatization, poor mental health outcomes, and the resultant substance use disorders [12]. These are important factors to consider in addressing barriers to the access and utilization of HIV care services. A recent study by Logie et al. (2017) demonstrated that stigmatization can make an individual increasingly susceptible to acquiring an HIV infection. The cross-sectional survey of 173 WLWH in Ontario also identified an association between the prevalence of racism and HIV stigma experienced by an individual [13]. COVID-19 added an additional layer of stigma and stress for many ACB WLWH. The additional health and socioeconomic burden of COVID-19 may have impacted the ability of ACB WLWH to access, utilize, afford, and to be motivated to engage with HIV care services.

Access to care cannot be defined solely by the availability of care services [14]. Conceptualizing an individual’s ability to access care must include a thorough understanding of the relevance, appropriateness, and affordability of the care services available [14]. Access to care and utilization of care services are interrelated concepts because the fitness of care services determines how an individual may access and therefore utilize the care services available to them [14]. ACB WLWH are susceptible to numerous structural barriers that may prevent effective access and utilization of HIV care services. A study examining resiliency factors related to how Black and Latino people living with HIV (PLWH) dealt with the COVID-19 pandemic indicated that the majority of HIV care services switched from in-person delivery to virtual delivery and over 70 percent of participants had at least one appointment canceled [8]. Although this study reports a different clinical context and population, it is important to recognize that virtual care delivery and canceled appointments may prevent ACB WLWH from accessing and utilizing HIV care services [8]. Field notes published early in the COVID-19 pandemic revealed significant gaps in the delivery of HIV care services when evaluating primary care services [15]. Specifically, COVID-19 caused a disconnect in the delivery of interdisciplinary care services, physical distancing strained the patient–provider therapeutic relationship, and the impact of factors such as access to care services, geographic location, food security, and employment status was intensified [15].

In Canada, most HIV care services are covered through the provincial health plan; however, specialized HIV services are not available in all geographic regions and therefore may impose financial costs, such as travel costs, upon an individual. In addition, there are associated expenses of living with HIV, for example buying fresh food, which may influence the affordability of HIV care services. The average cost of living, not including rent, utilities, or transportation for one person in Vancouver, BC’s capital, is CAD 1365 [16]. The average cost for a one-bedroom unit in BC is CAD 2471 [17]. The average total income for our sample was CAD 1862, significantly less than that necessary to live in the province of BC. An individual’s experience with being able to afford food, shelter, utilities, and clothing may influence their ability to engage with HIV care services. In a 2021 Canada-wide review of 15,556 ACB PLWH, 36 percent of participants reported a loss of work or income and 53 percent experienced food insecurity [10]. This review was conducted via a survey administered by the Public Health Agency of Canada. These results are consistent with the intensification of the impact of the social determinants of health for ACB WLWH, as well as other vulnerable populations.

The impact of the COVID-19 pandemic may have influenced the ability of ACB WLWH to access, utilize, afford, and to be motivated to engage with HIV care services. It is well established that the impacts of the COVID-19 pandemic disproportionately impacted ACB people [6,7]. Gwadz et al. (2021) revealed that the overall delivery of HIV care services was interrupted by the COVID-19 pandemic; however, PLWH in their study did not disengage with care [8]. This paper will explore how the COVID-19 pandemic influenced ACB WLWH and their ability to access, utilize, afford, and motivation to engage with HIV care services.

## 2. Materials and Methods

### 2.1. Design and Research Question

This study was a qualitative descriptive study [18]. We specifically addressed the following research question: What are the impacts of the COVID-19 pandemic on access, utilization, affordability, and motivation to engage with HIV care services for ACB WLWH in British Columbia (BC)?

### 2.2. Setting and Sample

This study was conducted in the province of British Columbia (BC), Canada. Eighteen women participated in this study who met the study’s inclusion criteria, i.e., they were at least 16 years of age, living with HIV for at least 3 months before the onset of the COVID-19 pandemic, and identified as African, Caribbean and/or Black. Both purposive and snowball sampling techniques were utilized to recruit participants from relevant ACB communities and HIV organizations in the province of BC. Participation was on voluntary basis.

### 2.3. Data Collection and Analysis

Data were collected through 60–90 min semi-structured interviews that were transcribed and analyzed by members of the research team. The interview questions pertained to the 4 major research questions in this study. Of note, some questions included: What are the perceptions of quality of life among ACB WLWH? What factors facilitate and impair quality of life among ACB WLWH based on the 4 domains of quality of life? What are the overall impacts of the COVID-19 pandemic on quality of life for ACB WLWH in BC, based on the 4 domains of quality of life? What are the overall impacts of the COVID-19 pandemic on access, utilization, affordability, and motivation to engage in HIV care services among ACB WLWH in BC?

Analysis was conducted by the primary investigator (E.O.) and research assistant (E.M.). E.O. and E.M. met regularly to discuss the findings and create a code book to categorize the data. Throughout the research process, the research team engaged in reflexive thinking to acknowledge how their experiences may have influenced their understanding and analysis of the research data. For example, after E.M. completed the initial transcription process, E.M. and E.O. engaged in lengthy discussions about the initial themes and compared this to any preconceived expectations of the dataset that were formed at the conclusion of the interviews. In-person and virtual interviews, depending on participant preference, were conducted between January and September of 2022. Prior to beginning the interview, participants who showed interest were screened for study eligibility. Informed consent and demographic information were then obtained. Data analysis was conducted through a thematic content analysis approach [19]. Member checking was conducted to ensure the trustworthiness and credibility of the data.

### 2.4. Ethical Considerations

The affiliated university at which this research was conducted granted ethical approval for this study. All results were kept confidential and only viewed by members of the research team. Data were securely stored in university-protected cloud software.

## 3. Results

### 3.1. Demographic Characteristics

Demographic characteristics are outlined in Table 1. Participants in this sample primarily accessed care at a specialized women’s health center (50%), were cisgender (94%), and heterosexual (100%). The average sample age was 47, ranging from 21 to 71. Participants had lived with HIV for an average of 14.1 years, ranging from 4 to 26. In subsequent sections, we outline the major findings of our study with direct examples from participants. All participants’ real names were changed to protect the confidentiality of the participants.

### 3.2. Access to HIV Services

Access to HIV care services was greatly impacted by the COVID-19 pandemic. All care services that could be delivered remotely were switched to online or telehealth delivery. The primary theme that emerged when asked about factors that influenced access to HIV care services for ACB WLWH was the delivery of impersonal or inadequate care services. The virtual delivery of care services made participants feel like they were not sufficiently assessed by their care provider. We present exemplar quotes from participants with their real names replaced with pseudonyms. For example, regarding access to care, Ellie commented that “sometimes your doctor can understand you better when they are actually seeing you”. Other participants felt that the doctors were personally turning them away: “you want to go see the doctor. And they’re gonna say no, we are not available. We will see people with the phone”—Angela.


*That day when the doctor told me? I think it was a receptionist, he said “go back home. We can’t help you, get that number [and] call [it]”. Yes, I’m a black, my English [is] not good. Why don’t [you] help me like a mother or like a sister. They didn’t help call. They didn’t help me. They just send me away, that is it.*
Maya


*It affected me in many ways, there was a time I was not feeling well. And I wanted to meet with the doctor, [but] it was not possible because they were only meeting with people on Zoom or [the] telephone for those who cannot come on Zoom. But I wanted to explain to them how I was feeling at that time but it was not possible.*
Lucy

Alternatively, some participants did not feel that their care services were impacted by their virtual delivery at all. Becky enthusiastically described the ease she experienced when seeking care:


*Well, it did not affect me badly. No, because as I told you, the hospital [name redacted], if [there is] anything they want to do for me [referring to medical procedures], they will just email me or text me or if they want for me to go and do tests, they will arrange everything for me whatsoever, they will just give me the dates or time, then I will go there … If they see any fault on me, they will call me straight to come and see how they can take care of that.*


Consistently amongst participants was the report that the delivery of support services dissolved quickly amongst the chaos and stress of the pandemic. Jessica described their disappointment at the lack of mental health services available: “Yeah, for mental health, we don’t get that anymore. Yeah, I wish we would just go back and have all those services”. Participants suggested that continuing outreach and support groups throughout future pandemics would help to improve quality of life and decrease experiences of isolation.

### 3.3. Utilization of HIV Care Services

The majority of participants did not feel that their ability to utilize HIV services was impacted by the COVID-19 pandemic. Sophie expressed that having ongoing care at a clinic prior to the start of the pandemic was a protective factor to encourage ongoing utilization of HIV care services: “I think the pandemic actually, like [didn’t] create any impact whether of utilizing because … the clinics that [you] have been assigned to, before the whole pandemic … So it didn’t really change anything”. A few participants felt dismissed by the decreased availability and support of healthcare providers and said that it decreased their utilization of care services overall.


*Yeah, it affected me at the time because I used to go to the clinic. And then now it was on Zoom. And again whenever you were sick between the course of time [the time of the pandemic] you couldn’t even see somebody because nurses and doctors were busy in the hospital attending to the Covid cases.*


A few participants recognized that there were challenges associated with seeing healthcare providers less frequently, such as running out of medication: “I wish they had like a system where they can see, oh, this person is almost running low. They need a refill, and then they will fill before I forget. Or maybe the pharmacy can have that. That would be awesome.”—Quinn. The only suggestion to improve utilization was to provide options of care delivery, such as taking the necessary precautions to facilitate in-person care services.

### 3.4. Affordability of HIV Care Services

In the context of BC healthcare services, most costs associated with HIV care are completely covered by the provincial government. Some participants did not feel that there were any extra expenses incurred related to HIV care services because of the COVID-19 pandemic. Ellie suggested that these expenses were present prior to the onset of the COVID-19 pandemic:


*I don’t think that COVID has affected it, it’s just people that are on disability, you are limited as to how much you can earn. And then therefore, you might not be able to pay for out of pocket services or medications. Because not all medications are covered by the healthcard.*


A few participants cited the cost of living and inflation of expenses as increased costs incurred during the pandemic. Eating a healthy diet with fresh fruits and vegetables or buying supplements can be a part of a patient with HIV’s healthcare regime that bears significant expenses for the individual: “the sum of COVID on the prices increased. So it’s a challenge.”—Francis.

One participant also shared that geography limits their access to HIV care, regardless of the COVID-19 pandemic, and that while some of the travel expenses are covered, many are not:


*If there was a service center that I could go to here on the island, but I would say that that’s just mainly for women like me or women living with HIV that can go and access services like that, or maybe I don’t know, support services, support groups like that, because I don’t think there’s any that I’ve come across here.*


Alicia also suggested diversifying the care services covered for patients with HIV: “it would be helpful for the like, other care services, like the dentist or the eye doctor to be covered as well”.

Factors to improve overall affordability should focus on coverage for extra expenses such as supplements and continuous funding support from government and organizations. Many participants noted that the grocery gift cards they received during the COVID-19 pandemic made a significant difference on their lives:


*It [COVID-19] was a blessing in disguise because we got funding. We were not going to get that funding anyway. We got funding, the emergency funding, it was a lot of money, people ended up being happy, having food. Even those who were not working, it was a good time, even those who lost jobs, they benefited a lot from the funding.*
Lucy

### 3.5. Motivation to Engage with HIV Care Services

Participants reported their motivation to engage with HIV care services was primarily deterred by fear. Fear of the COVID-19 pandemic, acquiring the virus, or the unknown impact of the virus on persons with decreased autoimmunity: “I was less likely to seek care because of COVID, because I was scared that whenever I go there, I take a risk to catch it.”—Rose


*You’d get scared of going to the hospital. You think when you go to Hospital you will fall sick of COVID … you’re scared to go. You’re always scared to go to the hospital.*
Jessica

One participant shared that their depression deterred their motivation to seek HIV care services: “No, I’m really bad. I’m not motivated at all … I think it’s just me being all alone, maybe not wanting to be around other people. It’s just like, it goes back to depression and you know, that I enjoy my own company.”—Alicia. Other participants either did not express similar fears or their motivation to seek care was not deterred by the impacts of the COVID-19 pandemic. Participants did not have any suggestions on how to improve motivation to engage with HIV care services in future pandemics.

### 3.6. Summary of Results

The results of the study are summarized in Table 2. The results were seperated into 2 categories: pre-existing factors and COVID-19 mediating factors.

## 4. Discussion

This study found that the COVID-19 pandemic had mixed effects on access, utilization, affordability of, and motivation to engage with HIV care services among ACB WLWH living in BC.

For some participants the COVID-19 pandemic negatively impacted access and utilization of HIV care services; for others, it made no difference. The absence of a difference in the care received was attributed to pre-pandemic assignment to a dedicated clinic. This gap was intensified for patients who were accessing primary care providers outside of their HIV specialist providers, and the additional difficulty of navigating the technological systems in place. This led participants to feeling unheard and/or estranged from their care services and providers. The onset and nature of the COVID-19 pandemic forcibly transitioned all non-essential healthcare delivery to virtual health formatting [20]. However, this finding highlights the fact that the ability to provide care in a virtual health capacity depends on having organized and resilient health systems that maintain quality of care for patients [21].

The undesirable effects of the pandemic were mediated by the transition to virtual care delivery, inadequate medication delivery and refill systems, shifted focus to COVID-19 cases, and collapse of support services such as mental health services. The switch to virtual care services was perceived to have hampered patient–provider relationships, particularly communication (absence of physical connection, patient feedback, attentiveness, and empathy from health providers). The shift to virtual care was brought about due to social distancing guidelines to protect patient’s safety. However, healthcare providers may have elected to continue using such distanced virtual practice even when the guidelines were no longer in place. Thus, participants felt unheard and/or estranged from their care services and providers. They had difficulty accessing primary care providers outside of their HIV specialist providers, and there was the additional difficulty of navigating the technological systems in place to utilize virtual health platforms. An important point in the pathway of care is the interaction of ACB WLWH with their provider [22]. Studies have shown that patient–provider interactions can pose barriers to access, utilization, and satisfaction with HIV care services for ACB patients in Canada [23]. A healthy and supportive patient–provider relationship is especially important because of the chronic nature of HIV, the need for optimum adherence (and viral suppression), and the intersectional stigma faced by ACB WLWH. The negative impact of the COVID-19 pandemic on access and utilization of HIV care services has been reported in systematic reviews. Most of the included studies also reflected the role of telemedicine in cushioning this effect [24,25,26]. Nonetheless, racial minorities such as ACB have been found to have lower preference for, satisfaction with, and associated health outcomes from virtual care delivery [27].

Participants found mental health services and social support groups useful because of the community and support provided. The removal of mental health services and social support groups worsened the isolation and loneliness associated with the pandemic control measures (lockdown, physical isolation, and limited movement). Mental health services are important for the health and wellbeing of ACB WLWH by improving overall utilization and adherence to HIV medications and services [28,29]. The intersection of the COVID-19 and HIV infection among WLWH could further worsen mental health outcomes and justify the prioritization of mental health services [30].

Alternatively, some participants preferred virtual care because of the seamlessness of conducting medical investigations and appointments. Similarly, another study of PLWH reported that they considered telemedicine care options important during the pandemic but this quantitative study did not assess their preferences for either virtual or in-person patient–provider appointments [4]. However, providers in this same study preferred telemedicine because it was better for patient engagement, patient-centered care, flexibility, and multidisciplinary engagement [4]. Interestingly, pre-pandemic studies showed a preference for telemedicine among PLWH because of reduced stress from transport and time wastage [31].

HIV care services are covered by the BC government. Therefore, the impact of the affordability of HIV care services primarily impacted the indirect costs of care. Indirect costs of care included transportation, food, supplements, and costs that were not covered by extended health plans. Given that ACB Canadians encounter disproportionate socio-economic disadvantages, additional HIV-related costs pose significant financial burdens for those already living below or close to the poverty line [32]. Likewise, similar studies have shown that the pandemic increased the costs of care for PLWH due to financial constraints, insurance costs, and increased personal expenses to comply with COVID-19 preventive measures [33]. These increased costs were often unattainable for this study’s sample of ACB WLWH, who were already struggling to meet their financial needs.

The government funding provided during the COVID-19 pandemic was useful in cushioning the financial impacts. During the pandemic, the Canadian government consistently paid pandemic relief benefits to Canadian residents [34]. This funding served as a social safety net for racial minorities such as ACB, especially for those with underlying and associated circumstances including job loss, under-employment, and financial challenges with increased costs of living during the pandemic [35].

The pandemic led to loss of motivation to use HIV care services, mostly due to psychosocial concerns. Participants expressed fears over contracting the COVID-19 virus while visiting health facilities. This fear was largely based on the knowledge of immunosuppression among those living with HIV. One of the participants noted that their depression deterred them from seeking care. Likewise, other studies showed that COVID-19 pandemic resulted in fear and low motivation to use health services and maintain adherence. This lack of motivation was due to low income and hunger, which took priority over adherence to appointments and medications [36,37]. Nevertheless, for some other participants, their motivation to use HIV care services was not affected by the pandemic.

In summary, the COVID-19 pandemic revealed significant health disparities and barriers to care for ACB WLWH and other marginalized populations. More research is needed into how support service delivery can be improved to increase access, utilization, affordability, and motivation to engage with HIV care services in future pandemics. Specifically, research is needed to understand how support and outreach groups can safely continue, and how care services can be delivered in a hybrid model that maintains the safety of all parties in future pandemics or in the occurrence of an event that disrupts healthcare services. In addition, support is needed to assist individuals in their community to work against the effects of systemic racism [8]. Gwadz et al. (2021) also suggested that measures to ensure equitable access to adequate telephone and internet services are essential to rely on virtual care delivery as an option for this population [8]. Nurses and other healthcare professionals should be aware that ACB WLWH may not have access to a phone or internet services in an environment where they can speak freely and may rely on the safe environment of a healthcare facility to conduct their care.

This study is not without limitations. Since the study was based on recall, experiences could have been under or over-reported. In addition, the COVID-19 pandemic was unprecedented, being the first pandemic since the HIV/AIDS pandemic, and largely negative, so participants could have felt more impacted by the negative effects.

## 5. Conclusions

This study found that the COVID-19 pandemic exerted a variety of effects on access, utilization, affordability, and motivation to use HIV care services for ACB WLWH. The virtual delivery of care services led some patients to feel dismissed by the healthcare system. While participants felt dismissed, they did not disengage with accessing HIV care services. Introducing a hybrid model of care could improve the access to and utilization and experience of HIV care services in future pandemics. Further research and support for a hybrid model of care that is satisfactory for sustaining patients’ intrinsic and extrinsic factors when dealing with HIV is necessary. In addition, it is essential that increased resources are dedicated to the maintenance and provision of mental health support for ACB WLWH. While most HIV care expenses are covered by provincial health plans, increased funding is needed to afford the range of healthy foods, supplements, and extended-health related expenses associated with the provision of holistic HIV care. Motivation to engage with HIV care services was deterred primarily by the fear of the COVID-19 virus. This fear was spurred by the misinformation that circulated throughout the COVID-19 pandemic and the unknown impact that the virus held for PLWH.

## Figures and Tables

**Table 1 ijerph-20-06051-t001:** Demographic information of study participants (N = 18).

Variable	Mean (M) or n	%
**Gender**		
Cisgender	n = 17	94
Transgender	n = 1	6
**Sexual Orientation**		
Heterosexual	n = 18	100
**Number of Years Living with HIV**	M = 14 Range = 4–26	
**Racial Identity**		
African	n = 10	55
Caribbean	n = 1	6
Black	n = 1	6
African/Black	n = 6	33
**Work Status**		
Full-time	n = 8	22
Part-time/Casual	n = 4	45
Unemployed	n = 6	33
**Income**	M = CAD 1862 Range = CAD 600–4000	
**Primary Location of Accessing HIV Care Services**		
Specialized Women and Children’s Community Health Centre	n = 9	50
Outpatient Interdisciplinary Services	n = 4	22
Hospital	n = 5	28
**Residency Status in Canada**		
Citizen	n = 6	33
Permanent Resident	n = 11	61
Student/Temporary Resident Status	n = 1	6
**Age**	M = 47 Range = 21–71	

**Table 2 ijerph-20-06051-t002:** Summary of Results.

Variable	Access	Utilization	Affordability	Motivation
Pre-existing factors (factors before the COVID-19 pandemic)	Effective communication by healthcare professionals	No changes	Direct costs of care covered by BC provincial health services Indirect health costs are not covered (i.e., dental, optometry)	No changes
COVID-19 mediating factors (factors intensified by the COVID-19 pandemic)	Telehealth/remote service delivery Impersonal/inadequate service delivery Removal of social support and mental health services	Dismissed by healthcare professionals Difficulty accessing medication refills	Inflation and increasing cost of living Travel expenses (to access care services) Funding/grocery store gift cards	Fear Misinformation about COVID-19 Depression

## Data Availability

Data are unavailable to maintain the confidentiality of participants involved.

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
