# Peer review of "How the COVID-19 Pandemic Influenced HIV Care: Are We Prepared Enough for Future Pandemics? An Assessment of Factors Influencing Access, Utilization, Affordability, and Motivation to Engage with HIV Services amongst African, Caribbean, and Black Women"

_ijerph, 2023, doi:10.3390/ijerph20116051_

Round 1

Reviewer 1 Report

Thank you for submitting this manuscript for consideration. I have enjoyed reading it but there are some amendments required prior to publication. 

Abstract: Written well and outlines paper.

Introduction - sets the scene well and provides an explanation for choosing the topic

Materials and Methods - This section needs to be improved. It is the weakest part of the paper at the moment. You state that this is a qualitative descriptive study but then comment on validity and reliability, which are more suited language for quantitative studies. It may have been better to comment on trustworthiness, credibility and transferability. How do we know that the findings are credible and trustworthy? You have not offered any discussion regarding reflexivity, an essential component of qualitative research, especially if there is an assumed power imbalance between researcher/interviewer and participant. It would have helped to produce a copy of your questions or prompts to illustrate what was asked. You have stated that the interviews were analysed by the researchers. Was this all six authors? How was agreement met? what happened with disagreements? 

Results - These are written well and are illustrated by the participants' voices. There is a range of voices, which is commendable. The themes are appropriate to the research question.  I am uncertain whether you have changed the names of the participants to protect confidentiality. I know some of these comments may sound self-explanatory but this will only be the case for those who are experienced and confident of research, so we need to ensure that we inform the reader appropriately. 

Discussion - your discussion appears to reflect the findings of the study. 

Conclusion - a satisfactory conclusion has been reached. 

Overall, this is a well-written paper. There are some minor typographical and grammatical errors that could be addressed with proof reading and copy editing.

Reviewer 2 Report

This study of self-reported access to HIV care during the COVID-19 pandemic by African, Caribbean, and Black women living with HIV (ACB WLWH) in BC, Canada is notable to help better understand the specific needs and experience of this marginalized population.

Abstract

1. Line 19: It is difficult to access whether 18 participants are sufficient to represent the diversity of the ACB WLWH population. This is compounded with the next comment to better put in context the relative size of this group.

Introduction

2. Line 32: Can the authors provide proportions for ACB WLWH for BC, the study group, rather than or in addition to the province of Ontario?

2a. Line 40: Here you use 'Covid-19', elsewhere 'COVID-19' is used. Please ensure consistency.

3. Line 46: Please note the location or provide some geographical information around the Etienne et al. (2020) study.

3. Line 66: Consider including a few examples of the 'unique health disparities for ACB Americans'.

4. Line 67: From this passage 'stigmatization of Covid-19 on ACB WLWH' the reader may not draw out that because people living with HIV are immunocompromised, they are at increased risk of acquiring SARS-CoV-2 infection. Is this the intention, or is there another point that can be drawn out? In additiona, the authors note earlier 'Of the identified 48 factors associated with increased risk of critical cases of Covid-19, originating from sub- Saharan Africa and older age', however, the reader does not yet know whether the authors are focused on or including WLWH who are ABC are also of advanced age for eaxample.

4. Line 75: Please note they study year for this 'cross-75 sectional survey of 173 ACB WLWH in Ontario' piece.

5. Line 89: Adding to 'The additional burden of Covid-19 may have impacted...' that the burden included health and socioeconomic burden.

6. Lines 102-110: It should be addressed that moving from in-person to virtual care was to protect the patients.

7. Line 116: Note the currency for these costing values. Line 117, there is a space missing in '1bedroom', also, 1 should be spelled out. And average income is monthly? It may be misaligned using the cost of living for Vancouver and the average cost of rent for the entire province. Those not familiar with Canadian geography may not understand the implications of Vancouver, ie, the provincial capital and an especially costly city.

8. Line 120:  Perhaps the word 'ability' is more appropriate than 'motivation'?

9. Lines 120-122: Note what year which review was conducted and to which pareiod this was compared. As well as please note the number of participants.

10. Line 123:  It may seem more balanced if it is acknowledged that the COVID-19 pandemic intensified the impact of various social determinants of health for ACB WLWH and well as other vulnerable populations.

11. Lines 124-126 and lines 130-131 are repetitive.

Materials and Methods

12. Line 136: 'engage in HIV care services' does not sound right to me. You can engage in care, or access care services. I appreciate the questionnaire cannot be written, but perhaps this can be reworded in the rest of the study where applicable.

13. Line 142: Could participants not identify as both for example African and Black (vs 'or')? As shown in table 1 'Africa/Black'.

14. Lines 156-158: Were participants anonymized (more than just replacing with pseudonyms)?

RESULTS

15. Table 1: Add '(M)' after 'Mean' in the header. Use the same number of decimal places for the mean value (14.1) and range (4-26). Same for income and age. For income, use '$' consistently and remove the extra space after the dash in the range value. Can a participant be both a citizen and a student? Or do the authors mean the participant has student resident status?

16. Lines 172: It is important to capture the experience of the participants. However, it is also important to again state that people were turned away from in-person care to protect the participant themself. Perhaps the messaging was not made clear as to why the shift in the mode of care, and that it was not a personal choice from the provider. If this is the case, this was a failing of the healthcare providers. But it should be brought out in my opinion. Also the extra strain the healthcare system was facing. Rationale for why mental health services were interrupted (lines 200+) -- this may later be addressed on line 313.

17. Line 213: Decreased utilization over what period, eg, from the beginning of the pandemic Q1 2022 until they were interviewed Jan-Sep 2022?

18. Lines 239-240: Is this statement true regardless of the pandemic? If HIV services were delivered virtually, other than ARV refill for example, wouldn't travel expenses decrease? The same applies to support services -- was this true during the pandemic? Would accessing in-person support services have been safe during all times during the pandemic. In general, some opportunities seem to be missed contextualizing these statements.

19. Additional provincial (or federal) policy to cover dental and optometry, supplements, grocery giftcards, care seem independent to the main objective for this study.

20. Table 2: left column, rows 1 and 2: 'Factors' does not need to be capitalized and for consistency.

DISCUSSION

21. A major lost opportunity is whether interruption in care caused an increase in HIV burden amount ACB WLWH in BC. It is acknowledged that there may be undereporting or underdetection.

22. Line 287: The statement 'without considering the option of hybrid care delivery', based on provincial mandates, would this have been an option?

23. Line 292: Can the authors speak to the reasons for inadequacy of medications, ie, supply chain issues, refill issues, stockouts, etc.?

24. Line 298:  The 'difficulty of navigating the technological systems in place', can the authors provide examples of reasons for this difficulty, ie, knowledge gap, technology gap (access to internet/Zoom, mobile phone), other reasons?

25. Line 316:  Consider replacing 'in' with 'among'. 

26. Line 317: I may be explicit, but it is clear that 'these' services are mental health services (vs HIV and COVID-related health services)?

27. Line 331: 'HIV related' is usually hyphenated.

28. Line 345: It seems 'when' or 'while' would be more appropriate than 'from'.

29. Line 346: I suggest replacing 'by HIV' with 'among those living with HIV'.

30. Lines 354-355: The authors should clarify whether how service delivery can be improved in general or during pandemic/outbreak situations.

31. Line 358: Would it be appropriate to expand this, in addition to in future pandemics, to other healthcare interruptions (eg, natural and other disasters or crises). 

32. Line 365: There is a first point ('Firstly') but no second, etc. points. Revise.

33. Line 367: The COVID-19 pandemic (6.9M deaths) is not unprecedented in term of death toll (ie, Bubonic plague 200M, Smallpox 56M Spanish flu 40-50M, etc.). However, it is the first instance of a pandemic since the onset of the HIV/AIDS pandemic. Please clarify.

34. Line 372: Led 'some' patients to feel dismissed.

35. Line 374: Can the authors say with certainty that a hybrid model will be possible to operate in future pandemics? Perhaps noting that it should have the framework in place and be considered and evaluated.

36. Line 376: Again, it will be more contextualized if the authors first note if and why mental health service resources were withdrawn during the pandemic. That is to further describe how health services were interrupted.

Please see a few minor comments made above.
